Vegetation–soil–microbiota dynamics across a 50-year reconstructed grassland chronosequence on the Loess Plateau of China

Ma Yuanyuan 1 2
Shen Yan 1 3 myynczx@163.com
Jin Ling 1 3
Tian Yu 1 3
Ma Hongbin 1 3
Lan Jian 1 3
Fu Bingzhe 1 3
1 College of Forestry and Prataculture, Ningxia University , Yinchuan , China
2 Ningxia Rural Science and Technology Development Center , Yinchuan , China
3 Northern Yanchi Desert Steppe Observation and Research Station of Ningxia , Yanchi , China
Karabiniuk Mykola
Electronic publication date: 2024 Dec 20
Publication date: 2024
Volume: 12
Electronic Location ID: e18723
Received 2024 Apr 30; Accepted 2024 Nov 26
Copyright: © 2024 Ma et al.
Copyright year: 2024
Copyright holder: Ma et al.
License: This is an open access article distributed under the terms of the Creative Commons Attribution License, which permits unrestricted use, distribution, reproduction and adaptation in any medium and for any purpose provided that it is properly attributed. For attribution, the original author(s), title, publication source (PeerJ) and either DOI or URL of the article must be cited.
License URL: https://creativecommons.org/licenses/by/4.0/

Keywords: Alfalfa, Soil quality, Rhizosphere microbiota, Reconstructed grassland

Funding: National Natural Science Foundation of China 32260349 Key Research & Development Project of Ningxia Autonomous Region, China 2023BCF01022, 2023BCF01043 Natural Science Foundation of Ningxia Autonomous Region, China 2024AAC03413 The research was supported by the National Natural Science Foundation of China (32260349), the Key Research & Development Project of Ningxia Autonomous Region, China (2023BCF01022, 2023BCF01043) and the Natural Science Foundation of Ningxia Autonomous Region, China (2024AAC03413). The funders had no role in study design, data collection and analysis, decision to publish, or preparation of the manuscript.

==============================
Alfalfa (Medicago sativa L.) establishment is an effective strategy for grassland reconstruction in degraded ecosystems. However, the mechanisms underlying vegetation succession in reconstructed grasslands following alfalfa establishment remain elusive. In this study, we investigated vegetation community, soil quality and rhizosphere microbiota dynamics across a reconstructed grassland chronosequence in the loess region of Northwest China. A space-for-time substitution method was used to evaluate grassland vegetation coverage and alfalfa production performance in nine stands of different ages (1–50 years old). High-throughput sequencing was conducted to characterise rhizosphere microbial communities associated with alfalfa. The plant heights, yields and stem-to-leaf ratios of alfalfa all peaked in the 7-year-old stand and then decreased in older stands, with Stipa bungeana replacing alfalfa as the dominant species in the 50-year-old stand. Soil bulk density and major nutrient contents were highest in the artificial grassland (1–10 years). Soil enzyme activities (e.g., urease and sucrase) were enhanced in the transitional grassland (10–30 years), accompanied by enrichment of potentially beneficial microbial taxa (e.g., Actinobacteria and Mortierella) and functional fungi (e.g., saprotrophs and symbiotrophs) in the rhizosphere. Soil water content, total porosity and rhizosphere microbial diversity reached their maximum levels in the natural grassland (>30 years). The results indicate that alfalfa establishment alters soil structure and nutrient status in the short term, creating an optimal rhizosphere micro-environment. The improved soil conditions and rhizosphere microbiota are favourable for subsequent establishment of native grass species, leading to the formation of a stable semi-natural grasslands.

Introduction

Desertification is a form of land degradation occurring in drylands, and it is one of the global problems targeted under the United Nations Sustainable Development Goals (Sterk & Stoorvogel, 2020). In the Loess Plateau region of Northwest China, land desertification is linked to severe soil erosion caused by a combination of harsh natural, historical and human factors (Wang & Zhang, 1993). More than 50 years ago, much of this region was barren with no vegetation (Han, 2009). However, before that, the area was predominantly covered with grasses and shrubs like Leymus chinensis, Agropyron cristatum, Medicago sativa, Caragana korshinskii, Caragana microphylla and many other plant species which were well adapted to the semi-arid conditions (Sneath, 1998). The local government attempted to reconstruct the ecological environment of desertified lands by aerial sowing of trees and grasses, to restore biodiversity and natural ecosystems (Han, 2009; Zhou, Li & Yue, 2020). In particular, establishing alfalfa (Medicago sativa L.) as a pioneer species is considered to be one of the most effective ways for revegetation of desertified lands because it provides diverse ecological functions and yields huge economic value (Yuan et al., 2016; Ma et al., 2022).

Alfalfa is a perennial plant that contributes to the improvement of soil carbon sequestration (Yuan et al., 2016; Mu et al., 2016). As a leguminous herb, alfalfa is able to fix atmospheric nitrogen through symbiosis with bacteria of the genus Rhizobium (Ma et al., 2022; Liu et al., 2023). Alfalfa establishment can increase soil microbial abundance and change vegetation community structure (Liu et al., 2023). Since this practice speeds up the succession of grassland vegetation, it has been widely adopted to reconstruct degraded systems on the Loess Plateau (Li & Shao, 2005; Wang et al., 2022a). Alfalfa is also widely cultivated in this region as a high-quality forage crop due to its ability to generate significant economic value through high yields and its support of the locally developed livestock industry. However, the lifespan of alfalfa ranges from 4 to 8 years (Wang, Sun & Yu, 2008; Wang et al., 2022a). It remains elusive how vegetation communities and soil properties evolve over time following alfalfa establishment, and what changes occur in alfalfa plants and associated rhizosphere microbial communities with increasing stand age. Answering these questions can enhance our mechanistic understanding of vegetation succession in reconstructed grasslands and provide useful information to control alfalfa decline.

Previous studies in the Loess Plateau region or nearby areas have shown that different changes in soil physicochemical properties or enzyme activities occur depending on the number of years since alfalfa establishment (e.g., Li & Shao, 2006; Cao, Sun & Yang, 2003; Shi, Zhang & Gao, 2017). For example, soil pH and phosphorus content initially increased to their highest levels in a 9-year-old alfalfa stand and then decreased with increasing stand age. Soil water content peaked in a 4-year-old alfalfa stand, whereas the lowest availability of soil nitrogen and potassium emerged in a 6-year-old stand (Li & Shao, 2006). Furthermore, enzymes secreted by soil microorganisms are key factors in nutrient cycling and their activities serve as sensitive indicators of soil quality (Cao, Sun & Yang, 2003). It was found that alfalfa establishment led to an initial increase followed by decrease in both soil catalase and urease activities, accompanied by a downward trend in soil enzyme activities from the surface to deeper soils (Shi, Zhang & Gao, 2017).

From the microbiota perspective, alpha-diversity of rhizosphere bacterial communities associated with alfalfa increased notably following alfalfa establishment in the Loess Plateau (Geng, Huang & Huo, 2020). In contrast, the viable counts of rhizosphere actinobacteria, other bacteria, and fungi in alfalfa stands trended downward with increasing stand age, especially in deeper soils (Nan, Shi & Yu, 2016). The rhizosphere harbours diverse microorganisms, which participate in soil nutrient cycling and contribute to soil structure stability (Allison & Goulden, 2017). The structure and function of rhizosphere microbiota characterise micro-environmental evolution in the root zone (Wu et al., 2014). Accordingly, the vegetation communities, soil properties, and rhizosphere microbiota in reconstructed grasslands are anticipated to change dynamically and interplay closely following alfalfa establishment (Geng, Huang & Huo, 2020; Xu et al., 2023).

This study was conducted to clarify the process and mechanism of vegetation succession in desertified land after alfalfa establishment over a 50-year period. Age-related changes in alfalfa yield and rhizosphere microbiota were also explored to provide guidance on the development of feasible grassland management strategies for controlling alfalfa decline. We hypothesised that: (1) Alfalfa establishment could improve soil physicochemical properties and enhance specific enzyme activities, thereby facilitating the colonization of other natural plant species and accelerating vegetation succession; and (2) alfalfa yield would decline over the long term due to the changes in soil conditions and reshaping of rhizosphere microbial communities. To verify these hypotheses, we surveyed vegetation community characteristics and analysed alfalfa production performance, soil quality and rhizosphere microbial community structure across a 50-year chronosequence of reconstructed grasslands. The results of the present study could unveil the dynamics of vegetation succession in reconstructed grasslands following alfalfa establishment and present empirical evidence supporting the management of old alfalfa stands. Portions of this text were previously published as part of a preprint (Ma, Shen & Jin, 2023).

Materials and Methods

Study area and experimental setup

The study was conducted in Longde County (35°21′–35°47′N, 105°48′–106°15′E), Guyuan City, Ningxia Hui Autonomous Region, China. As part of the Loess Plateau, the study area has an elevation ranging from 1,900 to 2,500 m above sea level. It belongs to a typical temperate continental climate zone, with an annual sunshine duration of 2,200 h. The mean annual temperature of the study area is 7.6 °C and its frost-free period lasts 140–160 days. The mean annual precipitation here is ~433.6 mm (mainly in June–September), and the mean annual potential evaporation is 1,360.6 mm. The major soil type in this area is grey cinnamon soil. A space-for-time substitution method was used to select nine grassland stands with consistent elevation, slope and aspect across different ages (1, 5, 7, 10, 15, 20, 30, 40 and 50 years old). All stands were spaced ~500 m apart from each other and managed under the same practices, with alfalfa seeds sown once 1–50 years ago. In each stand, three random quadrats (1 m × 1 m each) were selected for field survey and sampling.

Vegetation surveys and plant production performance analysis

Field surveys were carried out in July–August 2021 to obtain the total vegetation coverage and the coverage of major plant species, including the dominant species (plant density > 5, importance value > 0.1) and associated species (plant density: 0–5, importance value: 0–0.1). The density and height of the most abundant species were also recorded. The importance values of major plant species were calculated as follows (Yang, Han & Li, 1996):

Plant importance value (%) = (relative density + relative coverage + relative height)/3 × 100, where relative coverage (%) = (coverage of a specific plant species/coverage of all plant species) × 100; relative density (%) = (density of a specific plant species/density of all plant species) × 100; and relative height (%) = (height of a given plant species/height of all plant species) × 100.

At the time of vegetation surveys, the numbers of alfalfa plants in three quadrats of each age group were counted to calculate their density. Twenty alfalfa plants per quadrat were randomly selected to measure their vertical height and the results were averaged. Subsequently, the above-ground parts of selected alfalfa plants were cut from the ground. Another 20 alfalfa plants per quadrat were randomly sampled and separated into stems and leaves. All plant samples were placed into sample bags and brought back to the laboratory. After their fresh weights were determined, the above-ground plant parts were oven-dried at 80 °C for 48 h, and the dry weights were determined as the above-ground biomass. The stem and leaf samples were deactivated at 105 °C for 1 h and then oven-dried at 70 °C for 48 h. The dry weights of stem and leaf samples were obtained and used to calculate the stem-to-leaf ratio.

Bulk soil sampling and physicochemical analysis

In each quadrat, three soil profiles were dug to the 100 cm depth using a shovel. To determine soil physicochemical properties, Samples were then collected at 20 cm intervals (0–20, 20–40, 40–60, 60–80 and 80–100 cm depths) with a 7.5 cm-diameter soil auger. Soil water content was directly measured in the field using a TDR portable soil moisture meter (Imko, Ettlingen, Germany). After removing large stones and leaf litter, the samples were brought back to the laboratory and air-dried. To measure soil bulk density, intact samples were collected from depths of 0–20, 20–40 and 40–60 cm using 5 cm diameter cutting rings and dried in an oven at 150 °C until constant weight. Field capacity was measured by flooding irrigation in an enclosure frame (Xu et al., 2008). Soil total porosity was measured using the cutting ring method (Lipiec & Ku, 2006). Soil pH measurement was performed using a digital pH meter (PHST-5; Ohaus, Pine Brook, NJ, USA).

Soil nutrient status in the 0–100 cm profile was evaluated using standard testing methods (Qiu et al., 2022). Briefly, an automatic Kjeldahl nitrogen analyser (K-360; Buchi, Flawil, Switzerland) was used to determine soil total nitrogen (TN) content in all samples. Soil available nitrogen (AN) analysis was conducted based on the alkaline hydrolysis–diffusion method. Molybdenum-antimony anti-spectrophotometry (UV-1600; Mapada, Shanghai, China) was adopted to determine soil total phosphorus (TP) and available phosphorus (AP) contents, after sample digestion with a HClO4–H2SO4 mixture and extraction with a NaHCO3 solution, respectively. Soil total potassium (TK) and available potassium (AK) contents were determined by flame photometry (HP6410; Inesa, Shanghai, China) using samples fused with NaOH and extracted with NH4OAc, respectively. Determination of soil total organic carbon (TOC) content was conducted using the method of potassium dichromate oxidation–external heating.

Rhizosphere soil sampling and biological analysis

Twenty alfalfa plants were randomly selected from each quadrat and soil profiles were dug to the 50 cm depth within 50 cm from the stem of each plant. Alfalfa roots were exposed as completely as possible, and the loose soil adhering to fibrous roots was removed with a soil knife. After gently shaking the fibrous roots, the remaining soil adhering to roots (i.e., rhizosphere soil) from the 0–20 and 20–40 cm depths was brushed off with a sterilized hairbrush and passed through a 20-mesh sterilized sieve. After removing impurities with sterilized forceps, rhizosphere soils from twenty plants at depths of 0–20 and 20–40 cm were thoroughly mixed, respectively, then placed in sterile bags and kept in an ice box for transport. Upon arrival at the laboratory, each sample was divided into two parts. One part was used for enzyme activity assays (Gianfreda & Ruggiero, 2006), and the other part was immediately frozen at −80 °C for later use in microbial community analyses.

The MN NucleoSpin 96 Soil Kit (Macherey-Nagel, Düren, Germany) was used to extract total genomic DNA from rhizosphere soil samples. After quality check, the extracted DNA served as template for PCR amplification. The PCR primers used were 338F (5′-ACTCCTACGGGGAGGCAGCAG-3′) and 806R (5′-GGACTACHVGGGTWTCTAAT-3′) for bacteria (Sun et al., 2013), and SSU0817F (5′-TTAGCATGGAATAATRRAATAGGA-3′) and 1196R (5′-TCTGGACCTGGTGAGTTTCC-3′) for fungi (Borneman & Hartin, 2000). PCR amplicons were purified, quantified and normalised to construct sequencing libraries. The qualified libraries were sequenced on an Illumina HiSeq 2500 platform (llumina Inc., San Diego, CA, USA) at Biomarker Technologies (Beijing, China). Sequence analysis was carried out according to the methods described in Sun et al. (2019), with operational taxonomic units (OTUs) clustered based on the sequence similarity cutoff at 97%. The sequencing data are available at the NCBI Gene Expression Omnibus database (https://www.ncbi.nlm.nih.gov/) under accession numbers PRJNA1102710 and PRJNA1102694.

Data analysis

The vegetation and soil data were processed using Microsoft Excel 2010 (Microsoft Corp., Redmond, WA, USA). Based on OTU abundance data, microbial alpha-diversity indices, i.e., Chao1, Simpson index, Shannon-Winner index and abundance-based coverage estimator (ACE) of rhizosphere bacterial and fungal communities were estimated using QIIME v1.8.0 (Caporaso, Kuczynski & Stombaugh, 2010). To determine the effect of stand age on soil physicochemical properties, enzyme activities and rhizosphere microbial diversity, one-way analysis of variance and multiple comparisons of means by Duncan’s test were performed using SPSS 17.0 (SPSS Inc., Chicago, IL, USA), with a P value < 0.05 indicating significant difference. Pearson correlation coefficients were used to evaluate the relationship of microbial diversity to soil quality variables (i.e., physicochemical properties and enzyme activities). Response matrix and the explanatory/predictor matrix of soil physicochemical properties (dependent) and enzyme activities (independent) was performed using Canoco 5.0 (Microcomputer Power, Ithaca, NY, USA). The influence of soil quality variables (independent) on vegetation characteristics (dependent) was also explored using Redundancy analysis (RDA). Graphs were created using Sigmaplot 15.0 (Systat Software Inc., San Jose, CA, USA). Bacterial and fungal OTUs were assigned to functional groups (guilds) using PICRUST2 (Douglas et al., 2020) and FUNGuild (Nguyen et al., 2015), respectively.

Results

Vegetation community characteristics and alfalfa production performance

Based on the plant diversity of vegetation and the importance value of alfalfa, the course of desertified land revegetation could be divided into three stages (Fig. 1). In the first stage (artificial grassland, 1–10 years), alfalfa had a plant density >95 plants/m2 and an importance value >0.4, representing the dominant plant species. In the second stage (transitional grassland, 10–30 years), the dominant plant species were alfalfa, Stipa bungeana and Poa annua, which were associated with Plantago asiatica and Sonchus oleraceus. In the third stage (natural grassland, >30 years), S. bungeana and Artemisia scoparia were predominant and accompanied by Taraxacum mongolicum and Echinops gmelini. In the 50-year-old stand, alfalfa was completely replaced by S. bungeana. Table 1 summarises the dominant vegetation species, their plant densities and importance values in reconstructed grasslands of different age groups. Notably, plant heights, dry and fresh weights and stem–to–leaf ratios of alfalfa all peaked in the 7-year-old stand (Table S1).

Figure 1 Photographs of vegetation communities across a reconstructed grassland chronosequence in the loess hilly region of Northwest China (Longde County, Ningxia).

Table 1 Major vegetation species and their importance values in reconstructed grassland stands with different ages.

Stand age (year)	Dominant species	Plant density (plants/m2)	Importance value	Associated species	
1	Medicago sativa	97	0.53	Chenopodium aristatum	
Setaria viridis	27	0.13	Geranium wilfordii	
Chenopodium glaucum	14	0.15	Convolvulus arvensis	
5	Medicago sativa	100	0.36	Plantago asiatica	
Tripolium vulgare	88	0.23	Viola philippica	
Artemisia scoparia	86	0.15	Leymus secalinus	
7	Medicago sativa	113	0.42	Poa annua	
Artemisia scoparia	70	0.30	Plantago asiatica	
Agropyron cristatum	89	0.15	Saussurea japonica	
10	Medicago sativa	96	0.40	Viola philippica	
Artemisia scoparia	60	0.27	Cirsium japonicum	
Saussurea japonica	5	0.15	Plantago asiatica	
15	Medicago sativa	90	0.23	Plantago asiatica	
Poa annua	11	0.16	Ixeris chinensis	
Stipa bungeana	50	0.10	Artemisia scoparia	
20	Medicago sativa	30	0.22	Leymus secalinus	
Poa annua	20	0.24	Tripolium vulgare	
Stipa bungeana	7	0.14	Allium mongolicum	
30	Medicago sativa	9	0.14	Saussurea japonica	
Viola philippica	7	0.12	Convolvulus arvensis	
Stipa bungeana	26	0.60	Stipa breviflora	
40	Stipa bungeana	142	0.50	Heteropappus hispidus	
Medicago sativa	6	0.12	Echinops gmelini	
Artemisia scoparia	13	0.20	Agropyron cristatum	
50	Stipa bungeana	54	0.23	Tripolium vulgare	
Agropyron cristatum	6	0.10	Taraxacum mongolicum	
Stipa grandis	20	0.16	Heteropappus hispidus	

Soil physicochemical properties and rhizosphere enzyme activities

In the 0–100-m soil profile, soil bulk density was highest in the artificial grassland, whereas soil porosity and field capacity were both highest in the natural grassland. Soil water content varied with depth, being highest in the 0–20 cm soil layer of the transitional grassland, the 20–40 cm soil layer of artificial grassland and the 40–100 cm soil layer of natural grassland (Fig. 2). The soil pH did not change remarkably across different stand ages. Soil TOC, TN, TK and TP contents all trended downward over time, and their highest values were observed in the artificial grassland along the 0–100 cm soil profile. These soil nutrient contents were significantly higher in the 0–20 cm soil layer than in the deeper soil layers across different stand ages (Fig. 3). The mean values of rhizosphere urease, sucrase, phosphatase and catalase activities were generally higher in the transitional grassland than in the other two reconstruction stages (Fig. 4). RDA results revealed that TP, TN, TOC, TK and water contents as well as bulk density were significantly positively associated with rhizosphere catalase activity. TOC content was most closely associated with rhizosphere urease activity (Fig. 5A). Among the soil quality variables analyzed, soil pH and rhizosphere catalase activity were strongly associated with vegetation characteristics (Fig. 5B).

Figure 2 Comparison of soil physical properties at different stages.

BD, Bulk density; SP, Soil porosity; SW, Soil water content; FC, Field capacity.

Figure 3 Comparison of soil chemical properties at different stages.

TOC, Total organic carbon; TN, total nitrogen; TK, Total potassium; TP, Total phosphorus.

Figure 4 Soil enzyme activities in the rhizosphere of alfalfa with different stand ages.

(A) Urease activity, (B) sucrase activity, (C) phosphatase activity, and (D) catalase activity. All data are means ± standard deviation. Different letters near the error bars indicate significant difference among the age groups.

Figure 5 Redundancy analysis (RDA) plots showing the relationship of soil physicochemical properties to enzyme activities (A) and the relationship of soil quality to vegetation characteristics (B) in reconstructed grasslands.

a1: Alfalfa plant height; a2: Alfalfa stem diameter; a3: Alfalfa dry yield; a4: Number of species; a5: Alfalfa plant density; a6: Above-ground biomass; a7: Vegetation coverage; a8: Simpson dominance index; a9: Shannon Wiener diversity index; a10: Pielou evenness index.

Rhizosphere microbial community structure and predicted functions

A total of 4,210,659 (16S) and 4,319,972 (ITS) paired-end reads were obtained by high-throughput sequencing of 18 rhizosphere soil samples (nine stands and two depths). After quality control and assembly, 4,174,265 (16S) and 428,614 (ITS) clean reads were generated and then classified into 17,955 bacterial and 3,540 fungal OTUs, respectively. Among the different age groups, a total of 862 shared bacterial OTUs were identified in the 0–20 cm soil layer, with the largest number of unique bacterial OTUs (7) in the 20-year-old stand. In the 20–40 cm soil layer, there were 1,126 shared bacterial OTUs, and the number of unique bacterial OTUs (nine) was highest in the 10-year-old stand (Fig. S2A). Additionally, 164 shared fungal OTUs were identified in the 0–20 cm soil layer, with the largest number of unique fungal OTUs (14) in the 7-year-old stand. In the 20–40 cm soil layer, there were 212 shared fungal OTUs, with the largest number of unique fungal OTUs (14) in the 1- and 15-year-old stands (Fig. S2B).

The alpha-diversity of rhizosphere bacteria basically decreased first (artificial grassland) before increasing (transitional and natural grasslands) over time. Excluding ACE, the Chao1, Simpson, and Shannon-Winner indices of bacteria all peaked in the 40-year-old stand (Table S2). In contrast, the alpha-diversity of rhizosphere fungi initially increased (artificial grassland) and then decreased (transitional grassland) over time, followed by a late increase (natural grassland). Excluding Chao1, the Simpson, Shannon-Winner, and ACE indices of fungi were highest in the 50-year-old stand (Table S3). All alpha-diversity indices of bacteria and the Shannon-Winner and Simpson indices of fungi were higher in the 0–20 cm soil layer than in the 20–40 cm soil layer. At the phylum level, Proteobacteria (30–50%) was the most dominant bacterial group across all stand ages, especially in the 0–20 cm soil layer (Fig. 6A). The fungal community mainly comprised the phyla Ascomycola, Mortierellomycota and Basidiomycota (70%; Fig. 6C), with Mortierella, Dictyochaeta, Fusarium, Tetracladium, Apodus and Chaetomium (40%) as the predominant genera (Fig. S3). Interestingly, the relative abundance of Dictyochaeta increased towards older stands, whereas the other fungal genera had no distinct trends.

Figure 6 Relative abundances of major bacterial and fungal taxa (phylum level) at soil depths of 0–20 cm (A, C) and 20–40 cm (B, D) in the rhizosphere of alfalfa with different stand ages.

With regard to bacterial community functions, six categories at level 1 were predicted by PICRUSt2, and the most abundant functional category was metabolism (78.7–75.6%). At level 2, 10 pathways were identified, and the most abundant functional pathway was global and overview maps (42.5–41.7%). The bacterial community functions were not sensitive to stand age or soil depth (Fig. S4). The FUNGuild results showed that undefined saprotrophs (29.6–69.7%) were most predominant among the predicted fungal functional guilds. The fungal community functions varied remarkably with stand age, especially in the first 30 years. The relative abundance of saprotrophs was significantly higher than that of other guilds (P < 0.05) and peaked in the transitional grassland (P < 0.01). The relative abundance of symbiotrophs was also higher in the transitional grassland compared with that of other stages, albeit not significantly. In contrast, the relative abundance of pathotrophs slightly decreased in the course of vegetation succession (Fig. 7).

Figure 7 Predicted functions of rhizosphere fungal communities at different soil depths (A: 0–20 cm; B: 20–40 cm) and stages (C) in the reconstructed grassland.

Functional prediction was conducted using FUNGuild based on the top 10 functional guilds. *P < 0.05 and **P < 0.01.

Relationship of rhizosphere microbiota, soil quality and vegetation communities

Rhizosphere microbiota, soil quality and vegetation communities had mutual effects and influenced each other. The interplay between rhizosphere microbial diversity and soil quality was revealed by correlation analysis (Table S4). For the bacteria, the Simpson, Chao1 and Shannon-Winner indices were all positively correlated with TP content (P < 0.01). The bacterial Chao1 index was also positively correlated with AN and TOC contents (P < 0.05). For the fungi, the Chao1, ACE and Simpson indices were negatively correlated with soil water content (P < 0.05 or 0.01). The fungal ACE index was positively correlated with bulk density (P < 0.01), and the fungal Chao1 index was positively correlated with TP and AN contents (P < 0.05).

The RDA results unravelled the influence of soil quality on rhizosphere microbiota (Fig. 8). Bulk density, TP, AP and TOC contents were the leading factors influencing bacterial community structure at the phylum level (relative abundance >1%). Specifically, AP and TN contents were highly associated with Proteobacteria, bulk density was mainly associated with Gemmatimonadetes, and phosphatase activity was most closely associated with Bacteroidetes. Moreover, TP, AP, TN and AP contents were the key factors shaping fungal community structure at the phylum level. For example, TP, AP and AK contents were most closely associated with Glomeromycota and Basidiomycota, and urease activity was strongly associated with Zoopagomycota. Furthermore, pH and bulk density were prominently associated with Mortierellomycota and Chytridiomycota. At the genus level, the influence of soil quality on bacterial community structure diminished in grassland stands aged >20 years, and AP, TP and bulk density were the primary influencing factors. The influence of soil quality on fungal community structure diminished in stands aged >10 years, with AP, TP and TN being the most influential factors closely associated with Dictyochaeta and Thelonectria (Fig. 8D).

Figure 8 Relationship between rhizosphere bacterial (A, C) and fungal (B, D) community structure (at phylum and genus levels) and soil quality variables in reconstructed grasslands based on redundancy analysis (RDA).

M1: Urease; M2: Invertase; M3: Phosphatase; M4: Catalase; H1: Available phosphorus; H2: Total phosphorus; H3: Total nitrogen; H4: Available nitrogen; H5: Organic matter; H6: Available potassium; H7: Total potassium; H8: pH; H9: Water content; and H10: Bulk density.

The results indicated that specific rhizosphere microbial taxa of all the three stages (1–10 years, 10–30 years and >30 years) considerably influenced vegetation characteristics. Among the dominant bacterial phyla, Cyanobacteria and Gemmatimonadetes were most closely associated with alfalfa yield (dry weight). Actinobacteria was strongly associated with alfalfa density, and Firmicutes was chiefly associated with alfalfa coverage. Among the dominant fungal phyla, Chytridiomycota and Mortierellomycota were the most closely associated with alfalfa coverage and above-ground biomass, and Mucoromycota was principally associated with alfalfa stem diameter. At the genus level, Aeromicrobium and Vibrio (bacteria) were closely associated with alfalfa yield. Fusarium and Chaetomium (fungi) were strongly associated with plant species dominance, and Mortierella (fungi) was most closely associated with alfalfa yield (dry weight) and above-ground biomass (Fig. 9).

Figure 9 Relationship between rhizosphere bacterial (A, C) and fungal (B, D) community structure (at phylum and genus levels) and vegetation community characteristics in reconstructed grasslands of different ages, based on redundancy analysis (RDA).

a1: Alfalfa plant height; a2: Alfalfa stem diameter; a3: Alfalfa dry yield; a4: Number of vegetation species; a5: Alfalfa density; a6: Alfalfa above-ground biomass; a7: Vegetation coverage.

Discussion

The present study depicted long-term grassland reconstruction in desertified land after establishing alfalfa. Three distinct stages of vegetation succession were identified, namely, artificial grassland, transitional grassland and natural grassland. Following establishment, alfalfa showed the best production performance in the first stage (7-year-old stand). A long-term bottleneck in growing alfalfa is that biological decline occurs in old stands with increasing age. When growing for more than four consecutive years, alfalfa plants become smaller and sparser, with reduced plant height, decreased branch number, lower root activity, and increased disease incidence. This is a typical problem arising from continuous cropping of alfalfa, which leads to precipitous decline in crop yield (Rong, Shi & Sun, 2016; Wang et al., 2022b). The introduced alfalfa was ultimately replaced by native grass species in the 50-year-old stand, and this vegetation succession might be related to N2 fixation by alfalfa (Yu et al., 2018). The vegetation dynamics during long-term revegetation could be explained by associated changes in soil physicochemical properties and rhizosphere biological characteristics.

Soil bulk density and nutrient contents (e.g., TOC and NPK) were highest in the artificial grassland compared with the other two stages. Specifically, soil bulk density, AN and AP contents reached their maximum in the 1-year-old stand; TOC and TK peaked in the 5-year-old stand; the highest TP and TN contents were observed in the 7- and 10-year-old stands, respectively. This increase in soil nutrient contents was most likely a result of alfalfa root activity, which could pump phosphorus from the deep soil to the topsoil (Song et al., 2022) and fix atmospheric nitrogen (Yu et al., 2018). After alfalfa establishment for 10 years, TP and TN contents started to decrease, accompanied by alfalfa decline. These results indicate that soil structure was altered by establishing alfalfa and soil fertility was optimal during the early stage of grassland reconstruction. However, the greatest loss of soil water occurred in the transitional grassland year (40–100 cm depth), possibly because the extensive taproot system of alfalfa can take up water from deep soil layers and consume much more soil water than other forage grasses (Sim, Brown & Teixeira, 2017; Ren & Huang, 2016). The excessive loss of soil water might further contribute to the decline of alfalfa and the increased abundance of S. bungeana, as S. bungeana has less developed roots but higher drought resistance (Yang et al., 2019). This vegetation succession allowed soil porosity and water content continued to increase up to their maximum values, creating suitable growth conditions for other herbaceous plants. Based on the RDA results, there is a significant correlation between vegetation characteristics and soil pH, while TP and TN contents are primarily associated with alfalfa production performance. Accordingly, changes in these soil physicochemical properties could strongly influence the vegetation communities of reconstructed grasslands.

Soil enzymes are biocatalysts in soil biochemical processes and drive soil nutrient transformation (Li et al., 2021). Rhizosphere urease, sucrase, phosphatase and catalase activities all peaked in the transitional grassland among different stages of grassland reconstruction. This might be partly attributed to the colonization of S. bungeana, which can adapt to a relatively dry environment and enhance soil enzyme activities (Yang et al., 2019). The enhancement of soil enzyme activities could in turn contribute to the optimization of soil fertility. For example, urease is a key enzyme that determines nitrogen transformation in soil (Wang, Liu & Chen, 2021), and the highest urease activity we observed in the 10-year-old stand was consistent with the peak value of soil TN content. Additionally, the rhizosphere enzyme activities decreased from the 0–20 cm layer to the 20–40 cm soil layer. In the topsoil, favourable water and heat conditions coupled with high inputs of plant litter could facilitate enzyme production and activation. Both AP and TOC contents emerged as the key soil factors influencing rhizosphere enzyme activities. Notably, the catalase activity was positively correlated with soil nutrients (e.g., total NPK and TOC), soil water, bulk density and vegetation characteristics. These findings suggest that the vegetation characteristics of reconstructed grassland could alter soil physicochemical properties, consequently affecting rhizosphere enzyme activities (Eslaminejad, Heydari & Kakhki, 2020).

The species diversity of rhizosphere bacteria and fungi associated with alfalfa varied with stand age and both peaked in the natural grassland, most likely a result of high plant diversity at this stage. An increase in vegetation plant diversity can boost soil microbial diversity (Millard & Singh, 2009). Interestingly, there were no distinct trends in the composition of rhizosphere microbial communities, with Proteobacteria, Ascomycota, Mortierellomycota and Basidiomycota as the dominant taxa across different stand ages. The relative abundances of these dominant taxa were relatively low in the natural grassland among different stages. Different plant species at various grassland stages might shape rhizosphere microbial diversity by their distinct rooting strategies (Bakker, Mommer & van Ruijven, 2019; Bakker et al., 2021). With regard to soil depth, both the species diversity and taxa abundance of rhizosphere bacteria and fungi in the 0–20 cm soil layer were higher than those in the 20–40 cm soil layer. The loss of rhizosphere microbiota in diversity and abundance could be attributed to lack of plant litter and depletion of soil nutrients in the deeper layers. Soil TP and AP contents were the major soil factors shaping the structure of both bacterial and fungal communities in the rhizosphere of alfalfa. Similar results have been reported in a temperate steppe in Hulunbeir, Inner Mongolia (Liu, Fu & Zheng, 2010), and the arid zone of central Ningxia, China (Sun et al., 2019). However, our result is inconsistent with a previous finding by Smilauer (2001) that soil phosphorus content was not significantly correlated with rhizosphere fungal community composition or diversity in grassland. The discrepancy could be related to seasonal variability, environmental heterogeneity, and difference in host plants. Smilauer (2001) came to the conclusion based on only three grassland species (Achillea millefolium, Poa angustifolia and Plantago lanceolata) in an oligotrophic meadow of Czech at summer.

In the transitional grassland, Cyanobacteria, Actinobacteria, Mortierella and Glomus were enriched in the rhizosphere of alfalfa. Cyanobacteria can improve the structure of poor soils, thereby enhancing soil nutrient and water retention (Sepehr, Hassanzadeh & Rodriguez-Caballero, 2018). Actinobacteria exhibit high stress resistance in soil; they can synthesise antibiotics to inhibit plant pathogens and produce plant growth hormones to stimulate root growth, in addition to optimizing soil microbial community structure (Shirokikh & Shirokikh, 2019). Moreover, Cyanobacteria and Actinobacteria both comprise members capable of nitrogen fixation (Pajares & Bohannan, 2016). Furthermore, Mortierella is an abundant fungal genus in organic-rich soils and plays a key role in carbon and nutrient transformation (Uroz, Oger & Tisserand, 2016). Glomus spp. can form arbuscular mycorrhiza with terrestrial plants, and this symbiotic association facilitates the absorption of inorganic salts (especially phosphorus) in the soil by host plants (Li et al., 2008). Among the potentially beneficial bacteria, Cyanobacteria and Actinobacteria were the greatest contributor to alfalfa yield and density. As a group of potentially beneficial fungi, Mortierella mainly contributed to alfalfa yield and above-ground biomass. Moreover, predictive functional profiling by FUNGuild supported the dominant role of beneficial fungi in the transitional grassland, as the relative abundances both saprotrophs and symbiotrophs were highest at this stage. Saprotrophic fungi are involved in organic matter decomposition, nutrient mobilization, carbon cycling and soil structuring (Van der Wal et al., 2013). Symbiotrophic fungi confer host plants greater access to water and nutrients in exchange for carbon by vastly expanding the surface area of plant roots (Kramer et al., 2012). However, there is a limitation, as the fungal primers used here are not very effective at detecting arbuscular mycorrhizal fungi, which could lead to an underrepresentation of symbiotrophs. The rhizosphere microbial community variations in the transitional grassland could be related to the changing nutrient inputs into the rhizosphere by host plants, and the vegetation structure might be partly shaped by rhizosphere microbiota. A close association has been established between the succession of vegetation and rhizosphere microbial communities in a semiarid area (Song, Liu & Zhang, 2019). The enrichment of beneficial rhizosphere microbial taxa in the transitional grassland might be driven by the interspecific competition of alfalfa, which could release specific root exudates to recruit more beneficial microorganisms (Aschehoug et al., 2016). However, the competitiveness brought by beneficial rhizosphere microorganisms was reduced over time by various factors including soil moisture (Huang et al., 2018) and the lifespan of alfalfa (Wang et al., 2022a). As a consequence, alfalfa was ultimately replaced by native grass species (e.g., S. bungeana) with shorter roots and greater drought tolerance (Yang et al., 2019).

Conclusions

This study demonstrated vegetation successional dynamics in desertified land on the Loess Plateau after establishing alfalfa. We unveiled vegetation–soil–microbiota interactions in three stages of reconstructed grassland. During the early stage, soil nutrient status was improved over the short term, which was related to rhizosphere enzyme activities and conferred benefits to microbial community diversity in the artificial grassland. In the middle stage, the enhancement of enzyme activities in rhizosphere of alfalfa was accompanied by enrichment of potentially beneficial microorganisms and functional fungi, and the optimal micro-environment supported the growth of more dominant plant species in the transitional grassland. With continuous shifts in the soil conditions, alfalfa was eventually replaced by S. bungeana in the late stage, and a stable system was formed in the natural grassland. From an ecological conservation perspective, it would take at least 30 years to reconstruct barren land into a natural grassland through establishing alfalfa (aerial sowing) in the study area. For agricultural purpose, re-sowing of alfalfa is recommended for stand ages of ~10 years based on the highest alfalfa yield in the 7-year-old stand.

Supplemental Information

Supplemental Information 1 Study area in the loess hilly region of Northwest China (Longde County, Ningxia).

Supplemental Information 2 Venn diagrams showing the numbers of shared and unique operational taxonomic units of rhizosphere bacteria (A) and fungi (B) between different stand ages.

Different colors represents different stand ages.

Supplemental Information 3 Relative abundances of major bacterial and fungal taxa (genus level) at soil depths of 0–20 cm (A, C) and 20–40 cm (B, D) in the rhizosphere of alfalfa with different stand ages.

Supplemental Information 4 Composition of bacterial community functions was predicted by PICRUSt2 at level 1 (A) and level 2 (B).

a: 0–20 cm soil depth; b: 20–40 cm soil depth.

Supplemental Information 5 Alfalfa production performance in reconstructed grassland stands with different ages (Mean ± standard deviation).

Different letters in the same row indicate significant differences among the age groups (P < 0.05).

Supplemental Information 6 Changes in bacterial alpha-diversity in the rhizosphere of alfalfa with different stand ages.

Values followed by different letters are significantly different among the age groups or between the soil depths at P < 0.05.

Supplemental Information 7 Changes in fungal alpha-diversity in the rhizosphere of alfalfa with different stand ages.

Values followed by different letters are significantly different among the age groups or between the soil depths at P < 0.05.

Supplemental Information 8 Pearson correlation coefficients between rhizosphere microbial diversity indices and soil quality variables in reconstructed grasslands.

*P < 0.05 and ** P < 0.01.

Supplemental Information 9 Raw data of soil properties and soil enzyme activity.

Additional Information and Declarations

Competing Interests

Author Contributions

Data Availability

The authors declare that they have no competing interests.

Yuanyuan Ma conceived and designed the experiments, analyzed the data, prepared figures and/or tables, authored or reviewed drafts of the article, and approved the final draft.

Yan Shen conceived and designed the experiments, performed the experiments, authored or reviewed drafts of the article, and approved the final draft.

Ling Jin conceived and designed the experiments, performed the experiments, analyzed the data, prepared figures and/or tables, and approved the final draft.

Yu Tian analyzed the data, prepared figures and/or tables, and approved the final draft.

Hongbin Ma conceived and designed the experiments, authored or reviewed drafts of the article, and approved the final draft.

Jian Lan conceived and designed the experiments, authored or reviewed drafts of the article, and approved the final draft.

Bingzhe Fu conceived and designed the experiments, authored or reviewed drafts of the article, and approved the final draft.

The following information was supplied regarding data availability:

The data is available at NCBI GEO: PRJNA1102710, PRJNA1102694.

The raw measurements are available in the Supplemental Files.

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
