# Peer review of "Vegetation–soil–microbiota dynamics across a 50-year reconstructed grassland chronosequence on the Loess Plateau of China"

_PeerJ, doi:10.7717/peerj.18723_

## Round 0.1 · original submission · Major Revisions

The idea of the research and its subject matter are interesting, relevant and important for modern science. As a result of reviewing the manuscript, the reviewers expressed a significant number of comments regarding data processing, research methodology, substantiation of conclusions and other elements. Therefore, please read and respond, if possible, to the reviewers' comments and resubmit the manuscript.

Reviewer 1 ·

Basic reporting

The authors present a study on the vegetation composition, soil properties and rhizosphere microbial communities along a grassland restoration age gradient. The manuscript is clearly written, in good English and well structured. The number of abbreviations could be reduced to improve readability (e.g. BD, FC, SW, SP…). Most of the raw data is shared, except for the sequencing data. This type of sequencing data is normally submitted to databases like NCBI Genbank, but there is no mention of whether this has been done here.
The introduction lacks sufficient background to understand the goal of this study. The focus of this study is grassland restoration but how the degraded situation before restoration looks is unclear. Consequently, it is not clear how exactly alfalfa introduction helps restoration. The study lacks hypotheses or a clear goal. It is unclear to me why the authors of this study decided to look at the rhizosphere microbial communities of alfalfa and not to the bulk soil microbial communities or to the rhizosphere microbial communities of other plant species. After all, if the goal is to restore a native grassland, why did they not study the microbial communities associated with the plants that normally make up this grassland?
It is unclear how the authors expect the vegetation, soil properties and alfalfa rhizosphere microbes to change over the age gradient.
The captions of the figures do not sufficiently explain the figures. Fig. 2 and Fig. 3 appear to be two panels of one figure. So I would suggest to either take away the letters indicating the panels and adding a legend to Fig. 2 or to merge them into one figure.
Fig. 5 could be improved by showing the vegetation composition as points within the figure and by using more informative labels for the variables.
Fig. 6 could be improved by showing the different depths in separate figures because now it is quite hard to compare samples from the same depth since they are intermingled. The genus-level figures are also not very informative for the bacteria, so this could be excluded.
Fig. 7 A &B: there is basically no variation between samples so this figure does not really add much.
Fig. 7C There are quite some ectomycorrhizal fungi according to this figure. Are there ectomycorrhizal plants present in the grasslands? Could this be due to misclassifications? How did the authors deal with OTUs that were attributed to multiple classes, e.g. Ceratobasidium?

Experimental design

The authors used a space-for-time approach to study change in vegetation, soil properties and alfalfa rhizosphere microbes across an age gradient. However, no controls were included. It would be interesting to know how these restored conditions compare to the situation before alfalfa introduction and to a pristine native grassland that did not undergo restoration.
It is unclear how exactly alfalfa was introduced: was it seeded or planted? And whether microorganisms could have been co-introduced?
It is unclear which primers were used for sequencing. Phrasing suggest that the authors designed their own primers but the reasoning for this is unclear, since there already exist a large number of usable primers for this type of study. Primer sequences are not provided.

Validity of the findings

Given that there are no clear hypotheses or even a clear research goal, it is hard to assess the validity of the findings.
In the discussion, the authors infer causal relationships that cannot be inferred. For example: L 302-304: “The findings suggest that soil physicochemical properties could indirectly influence the vegetation characteristics of reconstructed grassland by altering rhizosphere enzyme activities.” In my opinion, it is more likely that the changing vegetation chararcteristics alter the soil physicochemical properties (that is the whole point of alfalfa introduction, right?), which subsequently affects the rhizosphere enzyme activities. Or it could simply be the vegetation that alters the rhizosphere microbial communities, which then affect the enzyme activities. From the correlations found in this study, no causal relationships can be established between soil properties, vegetation characteristics and enzyme activities.
The authors claim that they demonstrated that desertified land was restored, although they did not include a reference native grassland as a comparison. In my opinion, you cannot claim it was restored if we do not know what the native situation looks like.
They also claim that “the ecological mechanisms involved vegetation-soil-microbiota interactions”, but they only looked at microbiota of alfalfa and not of the native vegetation that they wanted to restore.
They claim that “alfalfa establishment is beneficial for mitigation of land desertification”, which might very well be true but given that they do not compare with a desertified grassland without alfalfa introduction, this cannot be concluded from their results.

Reviewer 2 ·

Basic reporting

In general terms, I appreciate the idea of this study. The space-for-time approach seems very appropriate and is a good solution to examine a wide range of restored grasslands. Linking soil properties, vegetation, and microbial communities makes the study more holistic, considering how variations in one variable might influence the trajectory of these grasslands.
However, the manuscript would benefit from a more comprehensive introduction and discussion, with additional details on the background of restoration in this region and explanations of the relationships between the variables. Emphasizing the ecosystem functions of alfalfa in these specific ecosystems and how these functions can enhance grassland restoration would add significant value. At times, the narrative lacks continuity, making it challenging to follow.
The English should be double-checked for accuracy. While I am not a native speaker, I do not feel I have the authority to comment on this.
Figures are incorrectly referred to in the text; their quality could be improved, and the font size increased.

Experimental design

The methods section should also be clarify (see comments in the attached file).

Validity of the findings

The data analysis methods are not clearly described. This section should be improved to enhance and allow the reproducibility of the analyses. Data from the rhizosphere microbiota was not present in the raw data.

Annotated reviews are not available for download in order to protect the identity of reviewers who chose to remain anonymous.

---

## Round 0.2 · Minor Revisions

The manuscript is significantly improved compared to the previous version, which indicates that the material has been prepared for publication. The main results of the study are relevant and will have a positive impact on the development of similar industry studies. After reviewing the submitted manuscript, the reviewers made minor comments regarding certain writing errors, the quality and detail of the labeling of the figures, etc which you should address.

Reviewer 1 ·

Basic reporting

The background information on the study area is much clearer in the revised version. However, I still have some questions. Using terms like ‘reconstruction’ suggests the aim is to go back to a certain vegetation type. So I think it would be good to mention the (natural?) vegetation that was present before desertification. Also there are several mentions of yield and economic value of alfalfa, yet there is no mention of why yield is important. Are these grasslands used for animal husbandry? Phrases like “stable natural ecosystem” suggests that the natural vegetation type (i.e. without human intervention) would be grassland. Is this the case? If the natural climax vegetation without human intervention (like animal husbandry) would rather be shrubland or forest I would rather call it semi-natural grasslands. It would also be good to mention the main reason for reconstructing the grasslands: is it mainly done to have grazing areas for animal husbandry or to restore biodiversity and natural ecosystems?

Experimental design

Unfortunately the fungal primers used here are not very good at picking up arbuscular mycorrhizal fungi. They will pick up some Glomeromycota but are strongly biased towards Ascomycota and Basidiomycota. This is an important limitation of the study as this causes an underrepresentation of this important group of symbiotrophs. This limitation should be mentioned in the discussion.

Validity of the findings

The authors should be more careful when concluding a cuasal link from the correlations they found. Some examples:
L315-320 Here you first suggest that the decrease in soil water content is due to the decrease in alfalfa but then you say that the decrease in soil water content causes alfalfa decline. This is contradictory. Here controls without alfalfa introduction (both desertified soil and ancient natural grassland) would have come in handy to try to disentangle cause and effect. Now you can only say that they are correlated but not which is driving which.
L322-323 Or the changing vegetation is affecting soil pH. Again, what you find is a correlation and which is driving which cannot be concluded from the results.
L323 But in the beginning you state that alfalfa is causing the increase and here you state that the increase in nutrients causes alfalfa productivity.

Additional comments

L165-166 the correct citation for this primer pair is Borneman & Hartin 2000
Borneman J, Hartin RJ. 2000. PCR primers that amplify fungal rRNA genes from environmental samples. Appl. Environ, Microbiol. 66: 4356–4360.

L361-362 These are also non-N-fixing plants, which could also contribute to a difference in the effect of soil phosphorous on their rhizosphere communities.

Reviewer 2 ·

Basic reporting

The manuscript has significantly improved from its initial version. The major comments provided by Reviewer 1 and myself have been thoughtfully addressed and implemented in a commendable manner. The readability of the text is now much smoother, clearer, and better structured. However, there are still a few areas, particularly regarding the clarity of the figures, where the manuscript would benefit from further refinement.

Experimental design

No comment

Validity of the findings

No comment

Additional comments

L. 184: In the section on Redundancy Analysis (RDA), it would be more precise to refer to the "Response matrix" and the "Explanatory/Predictor matrix." Additionally, in line 187, both soil quality variables and vegetation characteristics are categorized as “independent,” which may require clarification.

L. 228: I was unable to access the caption for Fig. S2. It is unclear what the different colors represent and how they relate to panels A and B. Please provide further clarification.

Figure 3: The caption contains a typo: “...properties at at different stages.” One of the instances of “at” should be removed.

Figure 9: The font size is very small, which hampers readability. Increasing the font size would greatly enhance the clarity of the figure. Additionally, panels A and C refer to bacteria, while panels B and D refer to fungi. However, it is unclear from which samples or grassland ages these data are derived (e.g., 0-10 years, 10-30 years, >30 years). This information should be made clearer either in the figure or in the corresponding paragraph (L. 281).

L. 222: In the section on rhizosphere microbial community structure and predicted functions, you mention “...18 rhizosphere soil samples (nine stands and two depths).” However, this contradicts the sampling method described earlier. In L. 151, you mention selecting 20 alfalfa plants from each quadrat, from two depths (0-20 cm and 20-40 cm), across three quadrats per stand, for a total of 9 stands. This would result in 20 × 2 × 3 × 9 = 1,080 rhizosphere samples. If samples were pooled by stand and soil layer, this should be clearly stated, as the current description is not sufficiently clear.

Line 303: The citation of Fukami et al. (2015) is incorrect; the correct citation is Fukami (2015). However, the inclusion of this reference here is not entirely appropriate. While the text refers to the facilitative effect of alfalfa as an N2-fixing legume, Fukami's paper focuses on priority effects and their underlying mechanisms, which are outside the scope of this study. If this reference is removed from the text, it should also be deleted from the References section.

---

## Round 0.3 · accepted · Accept

The manuscript meets the requirements and level of the scientific journal PeerJ. Revision and resubmission of the manuscript has significantly improved its quality. Based on the positive conclusion of the reviewers, the manuscript can be published in the journal, which will contribute to the dissemination of interesting research results among the academic community and scientists.

Reviewer 1 ·

Basic reporting

Most of my comments have been resolved. I just have one small comment.

Experimental design

no comment

Validity of the findings

L332-334 The authors still claim that a reduction in soil water can contribute to the alfa alfa decline, which I still find a strange claim. If the alfa alfa has no problem colonizing the desertified soils, surely low water availability shouldn’t be a problem later on?

Reviewer 2 ·

Basic reporting

no comment

Experimental design

no comment

Validity of the findings

no comment

Additional comments

Considering the substantial changes introduced by the authors since the initial review, I consider that the article entitled ‘Vegetation-soil-microbiota dynamics across a 50-year grassland restoration chronosequence in the loess region of China’ meets publication standards and is highly relevant to its field.
I believe the authors have addressed all the critical points raised during the review process, and I have no further comments to add. In my opinion, the article is suitable for publication.